# Convergence Analysis of ODE Models for Accelerated First-Order Methods via Positive Semidefinite Kernels

**Jungbin Kim**
Seoul National University
kjb2952@snu.ac.kr

**Insoon Yang**[*]
Seoul National University
insoonyang@snu.ac.kr

## Abstract

We propose a novel methodology that systematically analyzes ordinary differential equation (ODE) models for first-order optimization methods by converting the task of proving convergence rates into verifying the positive semidefiniteness of specific Hilbert-Schmidt integral operators. Our approach is based on the performance estimation problems (PEP) introduced by Drori and Teboulle [8]. Unlike previous works on PEP, which rely on finite-dimensional linear algebra, we use tools from functional analysis. Using the proposed method, we establish convergence rates of various accelerated gradient flow models, some of which are new. As an immediate consequence of our framework, we show a correspondence between minimizing function values and minimizing gradient norms.

## 1 Introduction

We consider the following convex optimization problem:

$$\min_{x \in \mathbb{R}^d} \ f(x), \tag{1}$$

where $f : \mathbb{R}^d \to \mathbb{R}$ is a continuously differentiable ($\mu$-strongly) convex function. We assume that a minimizer $x^*$ exists. First-order methods, for example, gradient descent and Nesterov's accelerated gradient method, are popular in solving this problem due to their low cost per iteration and dimension-free oracle complexities. These methods can be analyzed by examining their limiting ODEs. For instance, the gradient descent $x_{k+1} = x_k - s\nabla f(x_k)$ corresponds to the *gradient flow* $\dot{X}(t) = -\nabla f(X(t))$. Building upon this idea, Su *et al.* [36] derived the limiting ODE for Nesterov's accelerated gradient method (AGM) [27] and analyzed its convergence rate, offering valuable insights into momentum-based algorithms. A common approach to establishing convergence rates of continuous-time ODE models involves using Lyapunov functions [23].

In this paper, we propose a generic framework that builds upon the performance estimation problem (PEP) presented in [8] for analyzing the convergence rates of ODE models for various accelerated first-order methods, including Nesterov's AGM. The proposed method is designed from the Lagrangian dual of a relaxed version of continuous-time PEP. Consequently, our framework transforms the task of proving convergence rate into verifying the positive semidefiniteness of a specific integral kernel. Moreover, our framework can also ensure the tightness of the resulting guarantee, meaning that the obtained convergence guarantee is optimal among all possible convergence rates that can be derived from weighted integrals of the inequalities employed in convergence proofs. Using the proposed framework, we confirm the convergence rates of existing ODE models and uncover those of new accelerated ODE models. In traditional convergence analysis of ODE models, it can be challenging to design appropriate Lyapunov functions.[2] However, in our framework, we only need to verify

---

[*]Corresponding author.

[2]There is a line of work focused on systematically finding Lyapunov functions, often with the assistance of computers. See Section 1.1 for details.

37th Conference on Neural Information Processing Systems (NeurIPS 2023).

the positive semidefiniteness of a specific integral kernel. This approach circumvents the need for Lyapunov function design, making our framework more straightforward for analyzing convergence rates.

In the discrete-time setting, the PEP framework has been extensively studied for its ability to systematically obtain tight convergence guarantees [42] and facilitate the design of new optimization methods [17, 18, 40]. However, analyzing PEP is typically regarded as challenging to comprehend due to the involvement of large matrices with complex expressions. In contrast, our framework utilizes integral kernels, which serve as a continuous-time counterpart to matrices. The computational process within our approach yields simpler outcomes. Consequently, our continuous-time PEP framework has the potential to offer valuable insights into the analysis of the discrete-time PEP, similar to how ODE models have helped designing and analyzing discrete-time methods in the literature [20, 47]. By bridging the gap between the continuous and discrete settings, our methodology enhances the understanding of the PEP framework.

## 1.1   Related work

**Continuous-time models for first-order methods.**   The investigation into the continuous-time limit of accelerated first-order methods began with the study of AGM ODE [36, 2, 3]. Since then, subsequent researches have explored various aspects of the ODE models. These include generalizations within the mirror descent setup [20], a broader family of dynamics derived using Lagrangian mechanics [47, 48, 19], high-resolution ODE models [34, 33], and continuized methods [9]. Systematic methodologies for finding Lyapunov functions were developed, including deriving them from Hamilton's equations [6] or dilated coordinate systems [37]. For obtaining accelerated discrete-time algorithms, several studies have applied discretization schemes, such as symplectic [4] and Runge–Kutta [49] schemes, to discretize accelerated ODE models. [32] showed that applying multi-step integration schemes to the gradient flow also yields accelerated algorithms. A particularly relevant study is [19], as they present the dynamics in the form of (4) using the *H-kernel*, which plays a crucial role in our analysis.

**Performance estimation problems.**   The idea of using performance estimation problems to analyze the convergence rate of optimization methods was first introduced by [8]. This concept was further refined by employing a convex interpolation argument in [42] and was applied to a wide range of settings in [5, 43, 44, 12, 11, 7, 16]. The idea of performance estimation has been used to construct Lyapunov functions in [41, 39, 25, 45]. In particular, [25] analyzes continuous-time ODE models. However, their methodology differs from ours, as they employed semidefinite programs of finite dimension. Another closely related approach is based on *integral quadratic constraints* (IQC) from control theory [24], which were used to analyze the convergence rate of first-order methods in [21]. The IQC framework has been further studied in [14, 15, 10, 22, 31]. One practical application of PEP and IQC is the design of novel algorithms by optimizing the convergence guarantees. Some notable examples include OGM [17], TMM [46], ITEM [40], and OGM-G [18].

## 2   Preliminaries and notations

In this section, we review some basic notions from functional analysis that will be used throughout the paper. For a more detailed treatment, we refer the reader to the textbooks [29, 30, 28].

**Function spaces.**   We denote the set of continuous functions from $[0, T]$ to $\mathbb{R}^d$ by $C([0, T]; \mathbb{R}^d)$ and the set of continuously differentiable functions from $[0, T]$ to $\mathbb{R}^d$ by $C^1([0, T]; \mathbb{R}^d)$. We define the space $L^2([0, T]; \mathbb{R}^d)$ as the set of all measurable functions $f : [0, T] \to \mathbb{R}^d$ that satisfy $\int_0^T \|f(x)\|_{\mathbb{R}^d}^2 \, dx < \infty$. Then, $L^2([0, T]; \mathbb{R}^d)$ is a Hilbert space, equipped with an inner product and a norm defined by $\langle f, g \rangle_{L^2([0,T];\mathbb{R}^d)} = \int_0^T \langle f(t), g(t) \rangle_{\mathbb{R}^d} \, dt$ and $\|f\|_{L^2([0,T];\mathbb{R}^d)} = \sqrt{\langle f, f \rangle_{L^2([0,T];\mathbb{R}^d)}}$.

**Integral operators.**   An integral operator is a linear operator that maps a function $f$ to another function $Kf$ given by

$$(Kf)(t) = \int_0^T k(t, \tau) f(\tau) \, d\tau, \tag{2}$$

where $k : [0, T]^2 \to \mathbb{R}$ is the associated integral kernel. Intuitively, an integral kernel can be seen as a continuous-time version of a matrix. A Hilbert-Schmidt kernel is an integral kernel $k$ that is square integrable, i.e., $k \in L^2([0, T]^2; \mathbb{R})$. When $k$ is a Hilbert-Schmidt kernel, the associated integral operator $K$ is a well-defined operator on $L^2([0, T]; \mathbb{R}^d)$, called a Hilbert-Schmidt integral operator. If a Hilbert-Schmidt kernel $k$ is symmetric, i.e., $k(t, \tau) = k(\tau, t)$ for all $t, \tau \in [0, T]$, then the associated operator is also symmetric in the sense that $\langle Kf, g \rangle = \langle f, Kg \rangle$ for all $f, g \in L^2([0, T]; \mathbb{R}^d)$. Throughout this paper, we will use the term 'kernel' to refer to a Hilbert-Schmidt kernel.

**Positive semidefinite kernels.** A symmetric operator $K$ on a Hilbert space is said to be positive semidefinite and denoted by $K \succeq 0$ if $\langle Kf, f \rangle \geq 0$ for all $f$. When a symmetric kernel $k$ is associated with a positive semidefinite operator $K$, i.e., $\int_0^T \int_0^T k(t, \tau) f(t) f(\tau) \, dt d\tau \geq 0$ for all $f \in L^2([0, T]; \mathbb{R})$, we say that the kernel $k$ is (integrally) positive semidefinite and denote it by $k \succeq 0$. For continuous kernels, positive semidefiniteness of $k$ is equivalent to the following condition: $\sum_{i=1}^n \sum_{j=1}^n c_i c_j k(t_i, t_j) \geq 0$ for any $t_1, \ldots, t_n \in [0, T]$ and $c_1, \ldots, c_n \in \mathbb{R}$, given $n \in \mathbb{N}$.

**Proposition 1.** *We summarize some basic properties of continuous positive semidefinite kernels:*

(a) *For $\alpha \in C([0, T]; \mathbb{R})$, the kernel $k(t, \tau) = \alpha(t)\alpha(\tau)$ is positive semidefinite.*

(b) *For $k_1, k_2 \succeq 0$, their product $k(t, \tau) = k_1(t, \tau)k_2(t, \tau)$ is positive semidefinite.*

(c) *For $k \succeq 0$, its anti-transpose $(t, \tau) \mapsto k(T - \tau, T - t)$ is also positive semidefinite.*

(d) *If $\alpha \in C^1([0, T]; \mathbb{R}_{\geq 0})$ is an increasing function on $[0, T]$, then the symmetric kernel $k$ defined as $k(t, \tau) = \alpha(\tau)$ for $t \geq \tau$ and $k(t, \tau) = \alpha(t)$ for $t \leq \tau$ is positive semidefinite.[3]*

(e) *For $k \succeq 0$, we have $k(t, t) \geq 0$ for all $t \in [0, T]$.*

## 3 Continuous PEP for minimizing objective function value

In this section, drawing inspiration from its discrete-time counterpart [8, 42], we propose a novel framework for analyzing the convergence rate of ODE models for first-order methods, called the *continuous-time performance estimation problem (Continuous PEP)*. To illustrate this framework, we use the accelerated gradient flow as an example. Detailed steps can be found in Appendix C. Su *et al.* [36] derived the limiting ODE of Nesterov's AGM [27] as follows:

$$\ddot{X} + \frac{3}{t}\dot{X} + \nabla f(X) = 0, \tag{AGM ODE}$$

with initial conditions $X(0) = x_0$ and $\dot{X}(0) = 0$. Suppose we want to establish a convergence guarantee of AGM ODE in the form of

$$f(X(T)) - f(x^*) \leq \rho\|x_0 - x^*\|^2. \tag{3}$$

Here, we observe that the constant $\rho$ can be seen as an upper bound of the performance of AGM ODE for the criterion $(f(X(T)) - f(x^*))/\|x_0 - x^*\|^2$. To formalize this idea, we introduce the following optimization problem, which seeks to find the worst-case performance of the given ODE model:

$$\max_{\substack{f \in \mathcal{F}_0(\mathbb{R}^d; \mathbb{R}) \\ X \in C^1([0, T]; \mathbb{R}^d)}} \frac{f(X(T)) - f(x^*)}{\|x_0 - x^*\|^2}$$

$$\text{subject to} \quad X \text{ is a solution to AGM ODE with } X(0) = x_0, \, \dot{X}(0) = 0 \tag{Exact PEP}$$

$$x^* \text{ is a minimizer of } f,$$

where $\mathcal{F}_\mu(\mathbb{R}^d; \mathbb{R})$ denotes the set of continuously differentiable $\mu$-strongly convex functions on $\mathbb{R}^d$. This problem is useful to analyze the convergence properties of ODE models because the optimal value val(Exact PEP) of Exact PEP directly provides the guarantee (3) with $\rho = $ val(Exact PEP) regardless of any particular choice of $f$.

---

[3]Proof sketch: The kernel $k(t, \tau)$ can be expressed as a weighted integral of positive semidefinite kernels as $k(t, \tau) = \alpha(0)\mathbf{1}_{[0,T]}(t)\mathbf{1}_{[0,T]}(\tau) + \int_0^T \dot{\alpha}(s)\mathbf{1}_{[s,T]}(t)\mathbf{1}_{[s,T]}(\tau) \, ds$.

## 3.1 Relaxation of PEP

Exact PEP is challenging to solve due to the presence of an unknown function $f$ as an optimization variable. To address this difficulty, we relax the constraint $f \in \mathcal{F}_0(\mathbb{R}^d; \mathbb{R})$ with a set of inequalities that are satisfied by $f \in \mathcal{F}_0(\mathbb{R}^d; \mathbb{R})$. Before that, we first note that AGM ODE can be expressed as the following continuous-time dynamical system (see [19]):

$$\dot{X}(t) = -\int_0^t H(t, \tau) \nabla f(X(\tau)) \, d\tau \tag{4}$$

by setting $H(t, \tau) = \tau^3/t^3$. Here, $H(t, \tau)$ is called the *H-kernel*. We introduce two functions, $\varphi : [0, T] \to \mathbb{R}$ and $\gamma : [0, T] \to \mathbb{R}^d$, defined as follows:

$$\varphi(t) = \frac{1}{\|x_0 - x^*\|^2} \left( f(X(t)) - f(x^*) \right), \quad \gamma(t) = \frac{1}{\|x_0 - x^*\|} \nabla f(X(t)).$$

Using the chain rule and the convexity of $f$, we can derive the following equality and inequality:

$$
\begin{aligned}
0 &= \dot{\varphi}(t) + \left\langle \gamma(t), \int_0^t H(t, \tau) \gamma(\tau) \, d\tau \right\rangle, \\
0 &\geq \varphi(t) + \left\langle \gamma(t), v + \int_0^t \int_\tau^t H(s, \tau) \gamma(\tau) \, ds \, d\tau \right\rangle,
\end{aligned}
\tag{5}
$$

where $v = (x^* - x_0)/\|x_0 - x^*\|$. We can now relax Exact PEP by replacing its constraints with the equality and inequality above, resulting in the following problem:

$$
\begin{aligned}
&\max_{\varphi, \gamma, v} \quad \varphi(T) \\
&\text{subject to} \quad \text{(5) holds for all } t \in (0, T).
\end{aligned}
\tag{Relaxed PEP}
$$

Since any feasible solution to Exact PEP can be transformed into a feasible solution to Relaxed PEP, we have $\mathrm{val}(\text{Relaxed PEP}) \geq \mathrm{val}(\text{Exact PEP})$. Therefore, the convergence guarantee (3) holds with $\rho = \mathrm{val}(\text{Relaxed PEP})$ when using the proposed relaxation.

## 3.2 Lagrangian dual of relaxed PEP

To obtain an upper bound on $\mathrm{val}(\text{Relaxed PEP})$, we use Lagrangian duality. We introduce two *Lagrange multipliers* $\lambda_1 \in C^1([0, T]; \mathbb{R})$ and $\lambda_2 \in C([0, T]; \mathbb{R}_{\geq 0})$, where we imposed certain regularity conditions, such as continuity and differentiability, to ensure that the dual problem is well-defined. We then define the Lagrangian function as

$$
\begin{aligned}
\mathcal{L}(\varphi, \gamma, v; \lambda_1, \lambda_2) = \varphi(T) &- \int_0^T \lambda_1(t) \left( \dot{\varphi}(t) + \left\langle \gamma(t), \int_0^t H(t, \tau) \gamma(\tau) \, d\tau \right\rangle \right) dt \\
&- \int_0^T \lambda_2(t) \left( \varphi(t) + \left\langle \gamma(t), v + \int_0^t \int_\tau^t H(s, \tau) \gamma(\tau) \, ds \, d\tau \right\rangle \right) dt.
\end{aligned}
$$

When expressed in terms of the inner products in function spaces, we have

$$
\begin{aligned}
\mathcal{L}(\varphi, \gamma, v; \lambda_1, \lambda_2) = \varphi(T) &- \langle \lambda_1, \dot{\varphi} \rangle_{L^2([0,T];\mathbb{R})} - \langle \lambda_2, \varphi \rangle_{L^2([0,T];\mathbb{R})} \\
&- \frac{1}{2} \langle K\gamma, \gamma \rangle_{L^2([0,T];\mathbb{R}^d)} - \langle \lambda_2(t)v, \gamma(t) \rangle_{L^2([0,T];\mathbb{R}^d)},
\end{aligned}
\tag{6}
$$

where $K$ is the Hilbert-Schmidt integral operator with the symmetric kernel $k$ defined by

$$k(t, \tau) = \lambda_1(t) H(t, \tau) + \lambda_2(t) \int_\tau^t H(s, \tau) \, ds, \quad t \geq \tau.$$

The dual function is defined as $\mathrm{Dual}(\lambda_1, \lambda_2) = \sup_{\varphi, \gamma, v} \mathcal{L}(\varphi, \gamma, v; \lambda_1, \lambda_2)$. By weak duality, we have $\mathrm{val}(\text{Relaxed PEP}) \leq \mathrm{Dual}(\lambda_1, \lambda_2)$ for any feasible dual solution $(\lambda_1, \lambda_2)$. After performing some computations, we obtain the following expression for the dual objective function (see Appendix C):

$$\mathrm{Dual}(\lambda_1, \lambda_2) = \begin{cases} \inf_{\nu \in (0, \infty)} \{ \nu : S_{\lambda_1, \lambda_2, \nu} \succeq 0 \} & \text{if } \lambda_1(0) = 0, \ \lambda_1(T) = 1 \ \dot{\lambda}_1(t) = \lambda_2(t) \\ \infty & \text{otherwise,} \end{cases} \tag{7}$$

where $S_{\lambda_1,\lambda_2,\nu}$ is a symmetric kernel on $[0,T]^2$ given by

$$S_{\lambda_1,\lambda_2,\nu}(t,\tau) = \nu\left(\lambda_1(t)H(t,\tau) + \lambda_2(t)\int_\tau^t H(s,\tau)\,ds\right) - \frac{1}{2}\lambda_2(t)\lambda_2(\tau), \quad t \geq \tau. \quad (8)$$

We refer to $S_{\lambda_1,\lambda_2,\nu}$ as the *PEP kernel*. In Appendix H.2, we show that (8) can be viewed as the continuous-time limit of the discrete-time PEP kernel presented in [8].

To describe our framework, given $\nu_{\text{feas}} \in (0,\infty)$, suppose that the PEP kernel $S_{\lambda_1,\lambda_2,\nu_{\text{feas}}}$ is positive semidefinite with appropriate multiplier functions $\lambda_1$ and $\lambda_2$. Then, $\nu_{\text{feas}}$ is a feasible solution of the minimization problem in (7), and thus $\mathrm{Dual}(\lambda_1,\lambda_2) \leq \nu_{\text{feas}}$. On the other hand, by weak duality, $\mathrm{val}(\text{Relaxed PEP}) \leq \inf \mathrm{Dual}(\lambda_1,\lambda_2)$. Therefore, we conclude that $\mathrm{val}(\text{Exact PEP}) \leq \mathrm{val}(\text{Relaxed PEP}) \leq \mathrm{Dual}(\lambda_1,\lambda_2) \leq \nu_{\text{feas}}$, which implies that the convergence guarantee (3) automatically holds with $\rho = \nu_{\text{feas}}$.

Using this approach, we can recover the known convergence guarantee for AGM ODE in [36].

**Proposition 2.** *AGM ODE achieves the convergence rate* (3) *with* $\rho = 2/T^2$.

*Proof.* By choosing the multiplier functions $\lambda_1(t) = t^2/T^2$ and $\lambda_2(t) = 2t/T^2$, we can compute the PEP kernel (8) as $S_{\lambda_1,\lambda_2,\nu}(t,\tau) = (\nu - \frac{2}{T^2})\frac{t\tau}{T^2}$. Since the kernel $(t,\tau) \mapsto t\tau$ is nonzero and positive semidefinite, we have $S_{\lambda_1,\lambda_2,\nu} \succeq 0$ if and only if $\nu \geq 2/T^2$. Thus, we obtain $\mathrm{Dual}(\lambda_1,\lambda_2) = 2/T^2$, which establishes the convergence guarantee (3) with $\rho = 2/T^2$. $\square$

**Remark 1.** *Furthermore, this convergence guarantee is optimal among all possible guarantees obtained through the weighted integral of* (5). *The optimality of this rate follows from the fact that* $(\lambda_1,\lambda_2)$ *is the optimal solution to the dual problem* $\min_{\lambda_1,\lambda_2} \mathrm{Dual}(\lambda_1,\lambda_2)$. *See Appendix H.1 for details.*

### 3.3 Applying continuous PEP to various accelerated gradient flows

Note that the proposed method is not dependent on the choice of the H-kernel $H(t,\tau)$. Thus, it can be applied to arbitrary dynamics represented in the form of (4). Furthermore, while we have focused on the non-strongly convex case ($\mu = 0$) so far, the following paragraph demonstrates that our method can handle strongly convex objective functions by using a reparametrization technique.[4]

**Reparametrization from $\mathcal{F}_\mu(\mathbb{R}^d;\mathbb{R})$ to $\mathcal{F}_0(\mathbb{R}^d;\mathbb{R})$.** Consider a $\mu$-strongly convex objective function $f$. Since the proposed method is tailored for non-strongly convex objective functions, we choose to work with the convex function $\hat{f}(x) := f(x) - \frac{\mu}{2}\|x - x_0\|^2$ rather than working directly with $f$. Accordingly, we consider the following alternative formulation for the dynamical system (4), which involves $\nabla\hat{f}$ instead of $\nabla f$:[5]

$$\dot{X}(t) = -\int_0^t H^F(t,\tau)\nabla\hat{f}(X(\tau))\,d\tau. \quad (9)$$

The following theorem offers a general result that can be used to establish convergence guarantees for dynamical systems of the form (9).

**Theorem 1.** *Let $\nu > 0$ and $\lambda^F \in C^1([0,T];\mathbb{R}_{\geq 0})$ such that $0 \leq \lambda^F(0) < 1$, $\lambda^F(T) = 1$, and $\dot{\lambda}^F(t) \geq 0$ for all $t \in (0,T)$. Then, any solution to the integro-differential equation (9) satisfies*

$$\tilde{f}(X(T)) - \tilde{f}(x^*) \leq \lambda^F(0)\left(\tilde{f}(x_0) - \tilde{f}(x^*)\right) + \nu\|x_0 - x^*\|^2,$$

*where $\tilde{f}(x) := f(x) - \frac{\mu}{2}\|x - x^*\|^2$, if the following PEP kernel is positive semidefinite:*

$$S^F(t,\tau) = \nu\left(\lambda^F(t)H^F(t,\tau) + \dot{\lambda}^F(t)\int_\tau^t H^F(s,\tau)\,ds\right) - 2\alpha^F(t)\alpha^F(\tau), \quad t \geq \tau, \quad (10)$$

*where $\alpha^F(t) = \frac{1}{2}\frac{d}{dt}\{\lambda^F(t)(1 - \mu\int_0^t\int_0^s H^F(s,\tau)\,d\tau ds)\}$.*

The proof of Theorem 1 is done by finding a dual feasible point to the PEP and can be found in Appendix C. Below, we establish convergence rates for various ODE models using Theorem 1.

---

[4]In discrete PEP literature, a similar reparametrization technique was employed in [40, 13].

[5]The equivalent representations (4) and (9) are in a one-to-one correspondence. See Appendix C.1 for details.

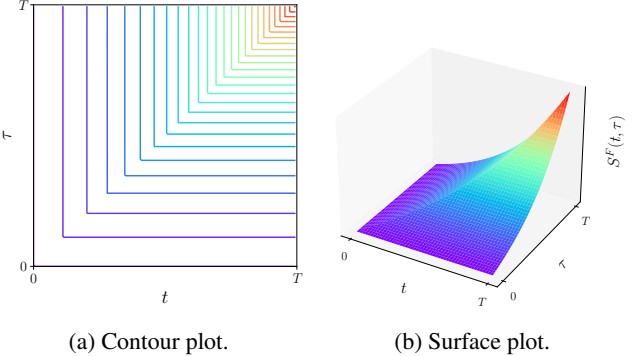

(a) Contour plot.        (b) Surface plot.

Figure 1: Visualization of the PEP kernel (11) for AGM-SC ODE.

**AGM-SC ODE.** We consider the following dynamical system modeling Nesterov's AGM for strongly convex case [26, Equation 2.2.22] (see [48, Equation 7]):

$$\ddot{X} + 2\sqrt{\mu}\dot{X} + \nabla f(X) = 0. \qquad \text{(AGM-SC ODE)}$$

This ODE model can be written as (9) with $H^F(t,\tau) = (1+\sqrt{\mu}\tau-\sqrt{\mu}t)e^{\sqrt{\mu}(\tau-t)}$ (see Appendix F.1). To use Theorem 1, we choose the multiplier function as $\lambda^F(t) = e^{\sqrt{\mu}(t-T)}$.[6] The PEP kernel (10) can be computed as (see Appendix G.1)

$$S^F(t,\tau) = \nu e^{\sqrt{\mu}(\tau-T)} - \frac{\mu}{2}e^{-2\sqrt{\mu}T}, \quad t \geq \tau. \qquad (11)$$

When $\nu = \frac{\mu}{2}e^{-\sqrt{\mu}T}$, this kernel is written as $S^F(t,\tau) = \frac{\mu}{2}e^{-2\sqrt{\mu}T}(e^{\sqrt{\mu}\tau} - 1)$, and is visualized in Figure 1. It is positive semidefinite since the function $\tau \mapsto e^{\sqrt{\mu}\tau} - 1$ is a nonnegative increasing function (see Proposition 1 (d)). It follows from Theorem 1 that AGM-SC ODE achieves the following convergence guarantee:

$$\tilde{f}(X(T)) - \tilde{f}(x^*) \leq e^{-\sqrt{\mu}T}\left(\tilde{f}(x_0) - \tilde{f}(x^*) + \frac{\mu}{2}\|x_0 - x^*\|^2\right), \qquad (12)$$

which is consistent with the well-known $O(e^{-\sqrt{\mu}T})$ convergence rate of AGM-SC ODE.

**Unified AGM ODE.** Using a unified Bregman Lagrangian framework, [19] obtained the following ODE that unifies AGM ODE and AGM-SC ODE:[7]

$$\ddot{X} + \frac{\sqrt{\mu}}{2}\left(\tanh_t +3\coth_t\right)\dot{X} + \nabla f(X) = 0, \qquad \text{(Unified AGM ODE)}$$

where $\tanh_t$ and $\coth_t$ denote the corresponding hyperbolic functions with the argument $\frac{\sqrt{\mu}}{2}t$. This ODE model can be written as (9) with $H^F(t,\tau) = (1 + \coth_t^2(\log(\text{sech}_t^2) - \log(\text{sech}_\tau^2)))\frac{\sinh_\tau \cosh_\tau}{\sinh_t \cosh_t}$ (see Appendix F.2). We select the multiplier function as $\lambda^F(t) = \sinh_t^2 / \sinh_T^2$. With this choice, the PEP kernel (10) can be expressed as follows (see Appendix G.2):

$$S^F(t,\tau) = \left(\nu - \frac{\mu}{2}\text{csch}_T^2\right)\frac{\tanh_t \tanh_\tau}{\sinh_T^2} + \nu\frac{\tanh_t \tanh_\tau \sinh_\tau^2}{\sinh_T^2}, \quad t \geq \tau. \qquad (13)$$

We show that this kernel is positive semidefinite for $\nu = \frac{\mu}{2}\text{csch}_T^2$. Proposition 1 (a) shows that the kernel $(t,\tau) \mapsto \tanh_t \tanh_\tau$ is positive semidefinite. Proposition 1 (d) shows that the kernel $(t,\tau) \mapsto \sinh_\tau^2$ is positive semidefinite because the function $\tau \mapsto \sinh_\tau^2$ is a nonnegative increasing function. Since the PEP kernel (13) with $\nu = \frac{\mu}{2}\text{csch}_T^2$ can be expressed as a product of two positive semidefinite kernels, it is positive semidefinite by Proposition 1 (b). Consequently, Theorem 1 implies that Unified AGM ODE achieves the following convergence guarantee:

$$\tilde{f}(X(T)) - \tilde{f}(x^*) \leq \frac{\mu}{2}\text{csch}_T^2\|x_0 - x^*\|^2. \qquad (14)$$

This guarantee aligns with the $O(\text{csch}_T^2)$ convergence rate reported in [19].

---

[6]As a rule of thumb, when the expected convergence rate is $O(\rho(T))$, we set $\lambda^F(t) = \rho(T)/\rho(t)$.

[7]This dynamical system models the unified AGM in [19] and the *constant step scheme I* [26, Equation 2.2.19].

**TMM ODE.** We consider the following novel limiting ODE for the *triple momentum method* (TMM) [46] (see Appendix E.1 for the derivation and a comparison with the one in [38]):

$$\ddot{X} + 3\sqrt{\mu}\dot{X} + 2\nabla f(X) = 0. \qquad \text{(TMM ODE)}$$

This ODE model can be written as (9) with $H^F(t, \tau) = -2e^{\sqrt{\mu}(\tau-t)} + 4e^{2\sqrt{\mu}(\tau-t)}$ (see Appendix F.3). By setting the multiplier function as $\lambda^F(t) = e^{2\sqrt{\mu}(t-T)}$, the PEP kernel (10) can be computed as (see Appendix G.3)

$$S^F(t, \tau) = 2\left(\nu - \mu e^{-2\sqrt{\mu}T}\right)e^{\sqrt{\mu}(t+\tau-2T)}, \qquad (15)$$

which is positive semidefinite for $\nu = \mu e^{-2\sqrt{\mu}T}$. Consequently, Theorem 1 implies that TMM ODE achieves the following convergence guarantee:

$$\tilde{f}(X(T)) - \tilde{f}(x^*) \le e^{-2\sqrt{\mu}T}\left(\tilde{f}(x_0) - \tilde{f}(x^*) + \mu\|x_0 - x^*\|^2\right), \qquad (16)$$

which is new to the literature. In Appendix H.3, we show that this convergence rate match aligns with the known convergence guarantee for TMM in the discrete-time case.

**ITEM ODE.** We consider the following new limiting ODE of the *information-theoretic exact method* (ITEM) [40] (see Appendix E.2 for the derivation of ITEM ODE):

$$\ddot{X} + 3\sqrt{\mu}\coth_t \dot{X} + 2\nabla f(X) = 0, \qquad \text{(ITEM ODE)}$$

where $\coth_t$ denotes the corresponding hyperbolic function with the argument $\sqrt{\mu}t$. This ODE model can be written as (9) with $H^F(t, \tau) = 4\sinh_\tau \cosh_\tau \coth_t \operatorname{csch}_t^2 + 2\sinh_\tau \operatorname{csch}_t(1 - 2\coth_t^2)$ (see Appendix F.4). By choosing the multiplier function as $\lambda^F(t) = \sinh^2(\sqrt{\mu}t)/\sinh^2(\sqrt{\mu}T)$, the PEP kernel (10) can be computed as (see Appendix G.4)

$$S^F(t, \tau) = 2\operatorname{csch}_T^2(\nu - \mu\operatorname{csch}_T^2)\sinh_t \sinh_\tau. \qquad (17)$$

For $\nu = \mu\operatorname{csch}_T^2$, this kernel is positive semidefinite. It follows from Theorem 1 that ITEM ODE achieves the following convergence guarantee:

$$\tilde{f}(X(T)) - \tilde{f}(x^*) \le \mu\operatorname{csch}_T^2\|x_0 - x^*\|^2, \qquad (18)$$

which is a novel result. In Appendix H.4, we show that this guarantee matches the known convergence rate for ITEM in the discrete-time case.

## 4 Continuous PEP for minimizing velocity and gradient norm

In this section, we present a result analogous to Theorem 1 to address convergence rates on the squared velocity norm $\|\dot{X}(T)\|^2$ or the squared gradient norm $\|\nabla f(X(T))\|^2$. For continuous-time ODE models, the analysis of convergence rates on the squared gradient norm $\|\nabla f(X(T))\|^2$ was first presented in [37]. However, their argument relies on the use of L'Hôpital's rule, which might give the impression that their approach is based on a clever trick or appears somewhat mysterious.

### 4.1 Translating convergence rates on $\|\dot{X}(t)\|^2$ into convergence rates on $\|\nabla f(X(T))\|^2$

In this subsection, we present a novel approach for establishing the convergence guarantee of ODE models on the squared gradient norm $\|\nabla f(X(T))\|^2$. The crucial insight lies in expressing $\nabla f(X(T))$ as $\int_0^T \nabla f(X(\tau))\delta_T(\tau)\,d\tau$, where $\delta_T$ denotes the Dirac delta function centered at $\tau = T$. Suppose we have a guarantee of the following form:

$$\left\|\int_0^T \alpha_t(\tau)\nabla f(X(\tau))\,d\tau\right\|^2 \le \rho\left(f(x_0) - f(x^*)\right), \qquad (19)$$

where $X \in C^1([0, T]; \mathbb{R}^d)$ and $\{\alpha_t\}$ is a family of functions parametrized by $t \in (0, T)$. In particular, we note that a convergence guarantee on $\|C(t)\dot{X}(t)\|^2$ of the dynamics (4) can be written as (19) with $\alpha_t(\tau) = C(t)H(t, \tau)$. A well-known argument for constructing the Dirac delta function (see

[35, Section 3.2]) shows that the weighted integral $\int_0^T \alpha_t(\tau)\nabla f(X(\tau))\,d\tau$ converges to $\nabla f(X(T))$ as $t \to T$, if the following conditions hold: $(i)$ $\alpha_t(\tau) \geq 0$, $(ii)$ $\int_0^T \alpha_t(\tau)\,d\tau \to 1$ as $t \to T$, and $(iii)$ for every $\eta \in (0, T)$, we have $\int_0^\eta \alpha_t(\tau)\,d\tau \to 0$ as $t \to T$. When $\alpha_t$ satisfies these properties, we say that the function $\alpha_t$ converges to the Dirac delta function $\delta_T$. Consequently, taking the limit $t \to T$ in (19) yields the following guarantee on $\|\nabla f(X(T))\|^2$:

$$\|\nabla f(X(T))\|^2 \leq \rho\left(f(x_0) - f(x^*)\right).$$

## 4.2 Convergence analysis via positive semidefinite kernels

In this subsection, we introduce a variant of continuous PEP that establishes the convergence rate on $\|\dot{X}(T)\|^2$ through checking the positive semidefininteness of the PEP kernel. Notably, this methodology can also prove convergence rates on $\|\nabla f(X(T))\|^2$ because the convergence rates on $\|\dot{X}(t)\|^2$ can be translated into those on $\|\nabla f(X(T))\|^2$, as discussed in the previous subsection.

**Reparametrization to time-varying functions.** In Section 3.3, we employed a reparametrization technique to deal with strongly convex objective functions. In this section, we first apply the same technique again, leading to the following expression:[8]

$$\dot{X}(t) = -\int_0^t H^G(t, \tau)\nabla\hat{f}(X(\tau))\,d\tau, \tag{20}$$

where $\hat{f}(x) := f(x) - \frac{\mu}{2}\|x - x_0\|^2$. However, we do not proceed directly with this form. Instead, we introduce an additional reparametrization step. Given a solution $X$ to (20), and a function $\lambda^G \in C^1([0, T); \mathbb{R}_{\geq 0})$, we define a family of functions $\{\hat{f}_t\}_{t \in [0,T)}$ as $\hat{f}_t(x) := \lambda^G(t)\hat{f}(x) - \langle\int_0^t \dot{\lambda}^G(\tau)\nabla\hat{f}(X(\tau))\,d\tau, x\rangle$. Then, we can show that (20) can be equivalently written in the following form (see Appendix D.1):

$$\dot{X}(t) = -\int_0^t \bar{H}^G(t, \tau)\nabla\hat{f}_\tau(X(\tau))\,d\tau, \tag{21}$$

for some kernel $\bar{H}^G$. The following theorem is analogous to Theorem 1 for our current purpose.

**Theorem 2.** *Let $\nu > 0$, $t_{\mathrm{end}} \in (0, T]$, $\alpha^G \in C([0, t_{\mathrm{end}}], \mathbb{R})$, and $\lambda^G \in C^1([0, t_{\mathrm{end}}]; \mathbb{R}_{\geq 0})$ such that $\lambda^G(0) = 1$ and $\dot{\lambda}^G(t) \geq 0$ for all $t$. Then, any solution to (21) satisfies*

$$\left\|\int_0^{t_{\mathrm{end}}} \alpha^G(\tau)\nabla\hat{f}_\tau(X(\tau))\,d\tau\right\|^2 \leq \nu \sup_{x \in \mathbb{R}^d}\left\{\hat{f}(x_0) - \hat{f}(x)\right\}, \tag{22}$$

*if the following PEP kernel defined on $[0, t_{\mathrm{end}}]^2$ is positive semidefinite:*

$$S^G(t, \tau) = \nu\bar{H}^G(t, \tau) - 2\alpha^G(t)\alpha^G(\tau), \quad t \geq \tau. \tag{23}$$

*In particular, the choice $\alpha^G(t) = C(t_{\mathrm{end}})\bar{H}^G(t_{\mathrm{end}}, t)$ gives a guarantee on $\|C(t_{\mathrm{end}})\dot{X}(t_{\mathrm{end}})\|^2$.*

The proof of Theorem 2 can be found in Appendix D. We now use this theorem to establish convergence rates of the anti-transposed dynamics[9] of the ODE models studied in Section 3.3.

**OGM-G ODE.** By taking the limit of the stepsize in OGM-G [18], Suh *et al.* [37] obtained the following ODE model for the non-strongly convex case ($\mu = 0$):[10]

$$\ddot{X} + \frac{3}{T - t}\dot{X} + \nabla f(X) = 0. \tag{OGM-G ODE}$$

This ODE model is the anti-transposed dynamics of AGM ODE, as it can be expressed as (20) with $H^G(t, \tau) = (T - t)^3/(T - \tau)^3$ (see Appendix F.5). To use Theorem 2, we choose $\lambda^G(t) = T^2/(T -$

---

[8]This expression is identical in form to (9), but we use the notation $H^G$ to avoid any notational overlap.

[9]We refer to (20) as the *anti-transposed dynamics* of (9), if $H^G(t, \tau) = H^F(T - \tau, T - t)$ for all $t, \tau$.

[10]We modified the coefficient of $\nabla f(X(t))$ from 2 to 1.

$t)^2$.[11] We set the terminal time $t_{\text{end}}$ before $T$ and apply a limiting argument to prove the convergence rate on $\|\nabla f(X(T))\|^2$. By setting $\alpha^G(t) = C(t_{\text{end}})\bar{H}^G(t_{\text{end}}, t)$ with $C(t_{\text{end}}) = 1/(T - t_{\text{end}})$, we can compute the PEP kernel (23) as (see Appendix G.5)

$$S^G(t, \tau) = \left(\nu - \frac{2}{T^2}\right) \frac{(T - t)(T - \tau)}{T^2}. \tag{24}$$

Since this kernel is the anti-transpose of the PEP kernel for AGM ODE in the proof of Proposition 2, we have $S^G(t, \tau) \succeq 0$ when $\nu = 2/T^2$ by Proposition 1 (c). By Theorem 2, we obtain the following inequality:

$$\left\|\int_0^{t_{\text{end}}} \frac{(T - t_{\text{end}})^2}{(T - \tau)^3} \nabla f(X(\tau)) \, d\tau\right\|^2 = \left\|\frac{\dot{X}(t_{\text{end}})}{T - t_{\text{end}}}\right\|^2 \leq \frac{2}{T^2} \sup_{x \in \mathbb{R}^d} \{f(x_0) - f(x)\}. \tag{25}$$

Observing that the function $\tau \mapsto \frac{(T - t_{\text{end}})^2}{(T - \tau)^3} \mathbf{1}_{[0, t_{\text{end}}]}(\tau)$ converges to $\frac{1}{2}\delta_T$ as $t_{\text{end}} \to T$, we have $\int_0^{t_{\text{end}}} \frac{(T - t_{\text{end}})^2}{(T - \tau)^3} \nabla f(X(\tau)) \to \frac{1}{2}\nabla f(X(T))$ as $t_{\text{end}} \to T$.[12] Substituting this result into (25), we deduce the following convergence guarantee:

$$\|\nabla f(X(T))\|^2 \leq \frac{8}{T^2} \sup_{x \in \mathbb{R}^d} \{f(x_0) - f(x)\}, \tag{26}$$

which recovers the known convergence rate of OGM-G ODE in [37].

**AGM-SC ODE.** We now analyze the convergence of AGM-SC ODE in terms of the squared velocity norm, using Theorem 2. This ODE model is the anti-transposed dynamics of itself, as it can be expressed as (20) with $H^G(t, \tau) = (1 + \sqrt{\mu}\tau - \sqrt{\mu}t)e^{\sqrt{\mu}(\tau - t)}$. We choose $\lambda^G(t) = e^{\sqrt{\mu}t}$ and $t_{\text{end}} = T$. By setting $\alpha^G(t) = C(T)\bar{H}^G(T, t)$ with $C(T) = \sqrt{\mu}/2$, the PEP kernel (23) is expressed as (see Appendix G.6)

$$S^G(t, \tau) = \nu e^{-\sqrt{\mu}t} - \frac{\mu}{2}e^{-2\sqrt{\mu}T}, \quad t \geq \tau. \tag{27}$$

Since this kernel is the anti-transpose of (11), it is positive semidefinite when $\nu = \frac{\mu}{2}e^{-\sqrt{\mu}T}$ by Proposition 1 (c). Therefore, we conclude that AGM-SC ODE achieves the following convergence guarantee:

$$\left\|\frac{\sqrt{\mu}}{2}\dot{X}(T)\right\|^2 \leq \frac{\mu}{2}e^{-\sqrt{\mu}T} \sup_{x \in \mathbb{R}^d} \left\{\hat{f}(x_0) - \hat{f}(x)\right\}, \tag{28}$$

which is new to the literature. A numerical experiment for this guarantee can be found in Appendix I.1.

**Unified AGM-G ODE.** In [19], the following unified AGM-G ODE is proposed:

$$\ddot{X} + \frac{\sqrt{\mu}}{2}\left(\tanh_{T-t} + 3\coth_{T-t}\right)\dot{X} + \nabla f(X) = 0, \qquad \text{(Unified AGM-G ODE)}$$

where $\tanh_{T-t}$ and $\coth_{T-t}$ denote the corresponding hyperbolic functions with the argument $\frac{\sqrt{\mu}}{2}(T - t)$. This ODE model is the anti-transposed dynamics of Unified AGM ODE, as it can be expressed as (20) with $H^G(t, \tau) = (1 + \coth_{T-\tau}^2(\log(\text{sech}_{T-\tau}^2) - \log(\text{sech}_{T-t}^2)))\frac{\sinh_{T-t}\cosh_{T-t}}{\sinh_{T-\tau}\cosh_{T-\tau}}$ (see Appendix F.6). To identify its convergence rate in terms of the squared gradient norm, we choose $\lambda^G(t) = \text{csch}_{T-t}^2 / \text{csch}_T^2$. By setting $\alpha^G(t) = C(t_{\text{end}})\bar{H}^G(t_{\text{end}}, t)$ with $C(t_{\text{end}}) = \frac{\sqrt{\mu}}{2}\text{sech}_{T-t_{\text{end}}}\text{csch}_{T-t_{\text{end}}}$, the PEP kernel (23) is expressed as (see Appendix G.7)

$$S^G(t, \tau) = \left(\nu - \frac{\mu}{2}\text{csch}_T^2\right)\frac{\tanh_{T-t}\tanh_{T-\tau}}{\sinh_T^2} + \nu\frac{\tanh_{T-t}\tanh_{T-\tau}\sinh_{T-t}^2}{\sinh_T^2}, \quad t \geq \tau. \tag{29}$$

Since this kernel is the anti-transpose of (13), it is positive semidefinite when $\nu = \frac{\mu}{2}\text{csch}_T^2$ by Proposition 1 (c). Therefore, Unified AGM-G ODE achieves the following convergence guarantee:

---

[11]As a rule of thumb, when the expected convergence rate is $O(\rho(T))$, we set $\lambda^G(t) = \rho(T - t)/\rho(T)$.

[12]In our argument using the Dirac delta function, we rely on the fact that the solution $X$ to OGM-G ODE can be continuously extended to $t = T$, which was shown in [37, Appendix D.3].

$$\left\| C(t_{\text{end}}) \dot{X}(t_{\text{end}}) \right\|^2 \le \frac{\mu}{2} \operatorname{csch}_T^2 \sup_{x \in \mathbb{R}^d} \left\{ \hat{f}(x_0) - \hat{f}(x) \right\}. \tag{30}$$

In Appendix G.7, we show that $C(t_{\text{end}})\dot{X}(t_{\text{end}}) \to -\frac{1}{2}\nabla f(X(T))$ as $t_{\text{end}} \to T$. As a result, we have the following convergence guarantee on $\|\nabla f(X(T))\|^2$:

$$\|\nabla f(X(T))\|^2 \le 2\mu \operatorname{csch}_T^2 \sup_{x \in \mathbb{R}^d} \left\{ \hat{f}(x_0) - \hat{f}(x) \right\}, \tag{31}$$

which recovers the known rate of Unified AGM-G ODE in [19].

The convergence analyses of the anti-transposed dynamics of TMM ODE and ITEM ODE are deferred to Appendix I.2. The convergence guarantees in this section can also be established using a Lyapunov function argument, as detailed in Appendix I.3.

## 5 Correspondence between minimizing function values and gradient norms

In Section 4.2, we verified the positive semidefiniteness of the PEP kernel (23) by showing its anti-transpose relationship with the PEP kernel (10), that is, $S^F(t,\tau) = S^G(T-\tau, T-t)$. The following proposition indicates that this relationship is not coincidental.

**Proposition 3.** *Suppose* $H^F(t,\tau) = H^G(T-\tau, T-t)$, $\lambda^F(t) = 1/\lambda^G(T-t)$, *and* $\alpha^F(t) = \alpha^G(T-t)$ *for all* $t$ *and* $\tau$. *Then, we have* $S^F(t,\tau) = S^G(T-\tau, T-t)$ *for all* $t$ *and* $\tau$.

The following proposition offers a general result for translating convergence rates on $\| \int_0^{t_{\text{end}}} \alpha^G(\tau) \nabla \hat{f}_\tau(X(\tau)) \, d\tau \|^2$ into those on $\|\nabla f(X(T))\|^2$.

**Proposition 4.** *Under the assumptions in Proposition 3, if* $\lambda^G(t) \to \infty$ *as* $t \to T$, *then we have* $\int_0^{t_{\text{end}}} \alpha^G(\tau) \nabla \hat{f}_\tau(X(\tau)) \, d\tau \to \frac{1}{2}\nabla f(X(T))$ *as* $t_{\text{end}} \to T$.

Both Propositions 3 and 4 have straightforward proofs through calculations, which can be found in Appendix J. Now, the next result naturally follows.

**Theorem 3.** *Under the assumptions in Propositions 3 and 4, the following statements are equivalent:*

(a) *Theorem 1 proves the convergence guarantee* $\tilde{f}(X(T)) - \tilde{f}(x^*) \le \nu \|x_0 - x^*\|^2$.

(b) *Theorem 2 proves the convergence guarantee* $\|\nabla f(X(T))\|^2 \le 4\nu \sup_x \{\hat{f}(x_0) - \hat{f}(x)\}$.

*Proof.* (a) holds $\underset{\text{Thm. 1}}{\Longleftrightarrow} S^F \succeq 0 \underset{\text{Prop. 3}}{\Longleftrightarrow} S^G \succeq 0 \underset{\text{Thm. 2, Prop. 4}}{\Longleftrightarrow}$ (b) holds. $\qquad\square$

## 6 Conclusion

In this paper, we have developed a novel and simple framework for analyzing the convergence rates of accelerated gradient flows via positive semidefinite kernels. Our framework enhances our understanding of continuous-time models for accelerated first-order methods and bridges the gap between the continuous-time and discrete-time PEP frameworks. For instance, the continuous PEP in Section 3 can be viewed as a continuous-time limit of the discrete PEP presented in [8] (see Appendix H.2). This connection unlocks new opportunities for studying PEP. Future research directions may involve extending our framework using techniques employed in the ODE models literature, such as generalizing to non-Euclidean settings [20, 1] or analyzing high-resolution ODE models [33].

**Codes.** The codes used to generate the figures in this paper are available at `https://github.com/jungbinkim1/Continuous-PEP`.

## Acknowledgments and Disclosure of Funding

This work was supported in part by the National Research Foundation of Korea funded by MSIT(2020R1C1C1009766), and the Information and Communications Technology Planning and Evaluation (IITP) grant funded by MSIT(2022-0-00124, 2022-0-00480).

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
