# OpenReview forum: "Convergence analysis of ODE models for accelerated first-order methods via positive semidefinite kernels"
_NeurIPS.cc/2023/Conference — NeurIPS 2023 poster_

### Official Review · Reviewer_48x5 · 2023-06-21

**Soundness:** 4 excellent
**Presentation:** 3 good
**Contribution:** 3 good
**Rating:** 6
**Confidence:** 4

**Summary:**

The article presents a continuous-time optimization framework for convex objectives which streamlines the process of giving rate guarantees.  Beginning from (Exact PEP), a difficult looking infinite-dimensional optimization problem.  This is relaxed using convexity, and then recast in a dual form, wherein it suffices to verify the positive definiteness of a PEP kernel.  This mirrors the discrete PEP kernel of Drori and Teboulle, and the authors show that this continuous PEP kernel can in fact be derived as a certain limit of the discrete one.

The article then deploys its method on a variety of accelerated ODE methods (Nesterov methods, triple momentum, information-theoretic exact methods, and others).  It additionally considers a variety of standard metrics, as appropriate for different situations (function value convergence, norm of gradient convergence).  The rates given appear to be consistent to state of the art  continuous-time guarantees or mirroring discrete-time guarantees

The method, which is designed to avoid construction of Lyapunov functions, instead requires construction of a positive definite kernel, which is built from the choice of a Lagrange multiplier function $\Lambda.$

**Strengths:**

1) The article is very clearly presented, with a large set of examples to illustrate the method.  This in particular important for illustrating the search for the magic $\Lambda.$
2) Certifying that the rate guarantee is correct, given the presented H-kernel (which can be computed directly from the ODE), and given the Lagrange multiplier is simple.
3) The article improves some existing rate guarantees and appears to reproduce many of the best-known guarantees, presumably found through more usual Lyapunov functional methods.  In this sense, the article identifies exactly the way in which these Lyapunov functional estimates are optimal (in the sense that the c-PEP guarantees produced this way follow (5) -- see Remark 1).
4) The method is extended systematically to give convergence of gradients (section 4).

**Weaknesses:**

1) While the paper removes the need to produce a Lyapunov function, it introduces the need to produce the Lagrange multiplier function $\Lambda$.  This does not appear to be systemized, and so it raises the obvious question to what extent it is necessary to learn the c-PEP machinery only to search for the $\Lambda$ when one could just search from the beginning for the more intuitive Lyapunov function.


**Questions:**

1) Is there a systematic way that you produce the Lagrange multiplier function? If so, this should be more prominently displayed.
2) The exact PEP is relaxed and them seems to play no further role.  Can you provide any insight in the degree to which this relaxation is sharp? (Are there examples where it is solvable?). Is there anything to say about Exact PEP at all, except for the relaxation?
3) Can you reverse engineer the c-PEP guarantee to produce optimization methods?
4) All the rate guarantees (for say the strongly convex case) look largely equivalent.  Is the best possible rate guarantee (say asymptotically) -- given a class of ODE methods with $a\sqrt{\mu}$ is second coefficient and $1$ its final coefficient $e^{-c(a)\sqrt{\mu}T}$?


**Limitations:**

The article appropriately addresses its limitations.

---

> ### Author Rebuttal · Authors · 2023-08-10
>
> We thank the reviewer for the positive feedback and thoughtful comments.
>
> > While the paper removes the need to produce a Lyapunov function, it introduces the need to produce the Lagrange multiplier function $\Lambda$. This does not appear to be systemized, and so it raises the obvious question to what extent it is necessary to learn the c-PEP machinery only to search $\Lambda$ for the when one could just search from the beginning for the more intuitive Lyapunov function.
>
> > Is there a systematic way that you produce the Lagrange multiplier function? If so, this should be more prominently displayed.
>
> While not explicitly mentioned in the paper, there is a rule of thumb for choosing $\Lambda$ (or $\lambda$). When the expected convergence rate is $O(1/a(T))$, we found that setting $\Lambda(t)=a(t)/a(T)$ in Theorem 1 (or $\lambda(t)=a(T)/a(T-t)$ in Theorem 2) leads to the desired results. For instance, in Line 165, because the known convergence rate of AGM-SC ODE is $O(1/e^{\sqrt{\mu}T})$, we set $\Lambda(t)=e^{\sqrt{\mu}t}/e^{\sqrt{\mu}T}=e^{\sqrt{\mu}(t-T)}$. In the final version of our work, we will include a formal explanation of this rule.
>
> > The exact PEP is relaxed and them seems to play no further role. Can you provide any insight in the degree to which this relaxation is sharp? (Are there examples where it is solvable?). Is there anything to say about Exact PEP at all, except for the relaxation?
>
> Dealing with the exact PEP itself is a highly challenging task, and to the best of our knowledge, there is no prior research exploring this direction, even in the discrete-time case. For the discrete PEP framework, [27] showed that using an interpolation argument, the exact PEP can be relaxed into a tractable form, and this relaxation is exact, meaning that the relaxed PEP is equivalent to the exact PEP. However, the relaxation technique applied in our continuous PEP framework mirrors the relaxation technique used for the discrete PEP in [2], which is known to be not tight (see [27, Section 1.4] for a related discussion). It could be an interesting future direction to find an exact tractable reformulation of the exact continuous PEP.
>
> > Can you reverse engineer the c-PEP guarantee to produce optimization methods?
>
> In the literature, there has been a line of work [7, 26] for producing optimal optimization methods by reverse engineering the discrete PEP presented in [2]. As our continuous PEP serves as a continuous-time counterpart of the discrete PEP, it can be used to produce optimal ODE models. Given a family of continuous-time models (4), parametrized by the kernel $H(t,\tau)$, we denote the best achievable convergence guarantee as $\mathrm{Guarantee}(H)$. The task of finding the optimal ODE model is then formulated as $\min_{H}\mathrm{Guarantee}(H)$. Using the continuous PEP, we can express $\mathrm{Guarantee}(H)$ as $\mathrm{Guarantee}(H)=\min_{\lambda}\mathrm{Dual}(H,\lambda)$, where $\mathrm{Dual}$ is the dual objective function defined in (7). Now, the task of finding the optimal ODE model can be formulated as $\min_{H,\lambda}\mathrm{Dual}(H,\lambda)$. One can generate the optimal ODE models by solving this problem, although it may not be a trivial task.
>
> > All the rate guarantees (for say the strongly convex case) look largely equivalent. Is the best possible rate guarantee (say asymptotically) -- given a class of ODE methods with $a\sqrt{\mu}$ is second coefficient and 1 its final coefficient $e^{-c(a)\sqrt{\mu}T}$?
>
> We can partially answer this question as follows: The ODE model $\ddot{X}+a\sqrt{\mu}\dot{X}+\nabla f(X)=0$ achieves a convergence rate of $O(e^{-(a-1)\sqrt{\mu}T})$, although we do not guarantee that this convergence rate is the best possible. Proof sketch: This ODE model is equivalent to the second Bregman Lagrangian flow in [30] with $\alpha(t)=\log\sqrt{\mu}$ and $\beta(t)=(a-1)\sqrt{\mu}t$. As a result, it achieves a convergence rate of $O(e^{-\beta(T)})=O(e^{-(a-1)\sqrt{\mu}T})$.

---

> > ### Comment · Area_Chair_fXPx · 2023-08-20
> >
> > Dear Reviewer,
> >
> > The author-reviewer discussion period is closing soon, so could you please go over the authors' rebuttal and respond with a message to the authors? It is important that authors receive a reply to their rebuttals, as they have tried to address comments raised by the reviewers.
> >
> > Best regards,
> > AC

---

### Official Review · Reviewer_PRzN · 2023-06-27

**Soundness:** 3 good
**Presentation:** 3 good
**Contribution:** 3 good
**Rating:** 7
**Confidence:** 3

**Summary:**

This paper proposes a novel methodology that analyzes ODE models for first-order optimization methods by converting the task of proving convergence rates into verifying the positive semidefiniteness of specific Hilbert-Schmidt integral operators. Based on the performance estimation problems (PEP) and functional analysis, the authors establish convergence rates of various accelerated gradient flow models.  The authors’ continuous time PEP framework provides insights into the analysis of the discrete-time PEP.

**Strengths:**

1. The authors developed a novel and simple framework for analyzing the convergence rate of continuous-time dynamics via positive semidefinite kernels.

2. The authors bridge the gap between the PEP framework for the continuous and discrete settings from continuous-time dynamics.

3. the authors’ continuous time PEP framework provides new opportunities for the analysis of the discrete-time PEP.


**Weaknesses:**

1. The results of this paper lack experimental validation.

2. The verification of the positive semidefiniteness of an integral kernel is difficult in implementation. How do you solve the difficulty in the calculation for the integral kernel?

3. There is a misspelling in line 39.


**Questions:**

See the weakness

---

> ### Author Rebuttal · Authors · 2023-08-10
>
> We thank the reviewer for the positive feedback and thoughtful comments.
>
> > The results of this paper lack experimental validation.
>
> In the initial submission of our paper, we did not include experimental validations, as our primary focus is to provide a new theoretical framework for convergence analysis of ODE models, rather than developing novel algorithms or ODE models. Additionally, most of the obtained convergence guarantee itself, or the corresponding discrete-time convergence guarantee, is already well-known in the literature. However, the convergence guarantee of AGM-SC ODE with respect to the measure $\Vert\dot{X}(T)\Vert^{2}$ is novel. We have performed experiments for this guarantee. The results can be found in Figure 3 of the attached PDF and will be included in the final version.
>
> > The verification of the positive semidefiniteness of an integral kernel is difficult in implementation. How do you solve the difficulty in the calculation for the integral kernel?
>
> We believe this concern does not significantly impact our contributions. Our work primarily focuses on establishing a theoretical foundation, rather than dealing with implementation. Due to the simplicity of our continuous PEP framework, compared to its discrete counterpart, all the results in our paper can be derived manually, without the need for numerical solvers.
>
> However, we agree that the implementation of continuous PEP could be a possible future direction. An integral kernel can be approximated with an arbitrarily small accuracy \epsilon by finite-rank integral operators (see Townsend & Trefethen, 2013), and one can readily verify the positive semidefiniteness of finite-rank operators. Thus, one can numerically verify the positive semidefiniteness of the given integral kernel, with an appropriate care of numerical approximation errors.
>
> > There is a misspelling in line 39.
>
> In Line 39, "typiccally" should be corrected to "typically". Thank you for the catch.
>
> ## Reference
>
> Townsend, A., & Trefethen, L. N. (2013). An extension of Chebfun to two dimensions. SIAM Journal on Scientific Computing, 35(6), C495-C518.

---

> > ### Comment · Reviewer_PRzN · 2023-08-17
> >
> > Thanks for your reply.  Your answers partially cleared my confusion. So I keep my rating.

---

### Official Review · Reviewer_RgYv · 2023-07-06

**Soundness:** 2 fair
**Presentation:** 3 good
**Contribution:** 2 fair
**Rating:** 5
**Confidence:** 4

**Summary:**

This paper presents a framework for analyzing convergence rates of a class of ODE models via the continuous-time performance estimation problem (PEP). The task of solving the PEP problem is relaxed into verifying the positive semidefiniteness of specific integral operators. The convergence rates of several accelarated gradient flow models were estabilished using the proposed method.

**Strengths:**

The novelty of the paper lies in the approach of defining and solving the continuous-time PEP problem to analyze ODE models. Under change of variables and Lagrangian duality, it is sufficient to verify the positive definiteness of specific integral operators. Using this method, the authors recover the known convergence guarantee and reveal unknown convergence rate for accelarated gradient flow models. The paper is clearly written and well-organized.

**Weaknesses:**

The intuition to consider the PEP kernel is not clear. The choices of multiplier function $\Lambda$ and $\nu$ are constructive and tricky for ODE models shown in section 3.3 (after Theorem 1) and section 4.2 (after Theorem 2). If given an ODE model as (4), it seems hard to distinguish if it can fulfill the assumptions in Theorem 1 and Theorem 2.

Lyapunov analysis for the discrete algorithms is much harder than the continuous dynamic system. The proposed approach only works for the continuous problems and didn't help the discrete case.

**Questions:**

There are some questions for technical unclarity.

- Does the paper consider only the strongly convex $f(x)$? If it is, then the minimizer $x^*$ is unique, I suggest the refer $x^*$ as 'the minimizer' rather than 'a minimizer' throughout the paper; if it is not, for convex $f(x)$, how to guarentee that the minimizer $x^*$ exists? For example, $f(x) = e^{-x}$. How to interpret the result of Theorem 1 in this case without defining $x^*$?

- For Exact PEP formulation in line 105, why the initial condition is set as $\dot X(0) =0$? If $x_0 = x^*$, then there is zero division issue, while this was not a problem in formulation (3). I suggest the authors adapt this case into the Exact PEP formulation.

- The computation stated in line 132 is not obvious. I suggest the authors include the outline in the paper.

- Suggest to outline the proofs of Theorem 1 and Theorem 2 in the paper.

- Can the convergence guarentee in line 155 imply the convergence guarantee of the ODE model? In what sense? For instance, as a extreme case, when $f(x) = \frac{\mu}{2}\|x - x^*\|^2$, $\tilde f(x) - \tilde f(x^*)$ is always zero, then Line 155 becomes trivial inequality.

- Is it possible that the supremum in the inequalities (15) is infinity and the inequality is trivial? For example, $\mu =  0$ and $f(x)$ is not bounded below.

- What is $x^T$ in equation (20) and (21)?

- Typos: line 45, 'enhances'; page 5, footnote 3 line 2, 'space'.

- Please check the format of the reference list, particularly the letter case (in [25], [27], [30], etc).

---

> ### Author Rebuttal · Authors · 2023-08-10
>
> We thank the reviewer for the thoughtful comments.
>
> > The intuition to consider the PEP kernel ...
>
> We want to clarify that selecting the multiplier functions is not such a challenging task. Although not explicitly mentioned in the paper, there is a rule of thumb for choosing $\Lambda$ (or $\lambda$). When the expected convergence rate is $O(1/a(T))$, we found that setting $\Lambda(t)=a(t)/a(T)$ in Theorem 1 (or $\lambda(t)=a(T)/a(T-t)$ in Theorem 2) leads to the desired results. For instance, in line 165, because the known convergence rate of AGM-SC ODE is $O(1/e^{\sqrt{\mu}T})$, we set $\Lambda(t)=e^{\sqrt{\mu}t}/e^{\sqrt{\mu}T}=e^{\sqrt{\mu}(t-T)}$. Once the appropriate $\Lambda$ (or $\lambda$) is determined, it is not difficult to select the parameter $\nu\in\mathbb{R}$ that makes the PEP kernel positive semidefinite. In the final version of our work, we will include a formal explanation of this rule.
>
> > If given an ODE model as (4), ...
>
> In our analysis, checking whether the given ODE model satisfies the assumptions in Theorem 1 and Theorem 2 is indeed straightforward. Once the multiplier function $\Lambda$ (or $\lambda$) is chosen, the verification process involves only computing the PEP kernel and checking if it is positive semidefinite.
>
> Regarding the soundness of assumptions, Theorem 2 in the initial submission of our paper relied on a nontrivial assumption (16). Fortunately, we have successfully relaxed this assumption, and the revised version will be included in the final version of our paper. In the general response, we provide a detailed explanation of the modification we made.
>
> > Lyapunov analysis for the discrete algorithms ...
>
> While it is true that discrete-time analysis can be more challenging than continuous-time analysis, we disagree that continuous-time analysis does not help discrete-time analysis. In fact, our continuous-time PEP framework can provide guidance for analyzing the discrete PEP. In the discrete PEP, finding an appropriate multiplier vector $\lambda_{i}$ can be quite challenging. However, this task becomes more approachable in continuous-time analysis, where we need to find a multiplier function $\lambda(t) $ for the corresponding continuous PEP. Once we have a multiplier function $\lambda(t)$ that works for the continuous PEP, we can discretize this $\lambda(t)$ to obtain candidates for multiplier vectors $\lambda_{i}$ that work for the discrete PEP (see Appendix G.2.2 for a specific example and related discussion). Furthermore, discretizing a multiplier function $\lambda(t)$ is typically simpler than discretizing a Lyapunov function. As a result, transitioning from continuous-time analysis into discrete-time analysis is more straightforward in the PEP framework than in traditional Lyapunov analysis.
>
> > Does the paper consider only the strongly convex ...
>
> In our paper, we assume the existence of a minimizer $x^*$ (we will make this clearer in the final version). This assumption is standard in the literature on the analysis of accelerated first-order methods and can be found in Nesterov's seminal paper on AGM [15] as well as recent works [22,19] published in NeurIPS.
>
> > For Exact PEP formulation in line 105, ...
>
> The initial condition $\dot{X}(0)=0$ is a part of the continuous-time model (AGM ODE), and its derivation can be found in Su et al.'s paper [23]. Note that without having an initial condition for both $X(0)$ and $\dot{X}(0)$, the solution to AGM ODE is not uniquely determined. In the final version, we will mention the initial condition immediately after introducing AGM ODE for the first time.
>
> > If $x_0=x^*$, then there is zero division issue, ...
>
> While it is desirable to avoid such an issue, we don't need to worry about this. The case $x_0=x^*$ leads to $X(t)=x^*$ for all $t$, making the situation so trivial that we can safely exclude this case from our analysis. Also, note that dividing by $\|x_0-x^*\|$ can be commonly found in the PEP literature [2,26].
>
> > Can the convergence guarentee in line 155 ...
>
> The convergence guarantee in line 155 is for $\tilde{f}(X(T))-\tilde{f}(x^*)=f(X(T))-f(x^*)-\frac{\mu}{2}\|X(T)-x^*\|^{2}$, rather than for $f(X(T))-f(x^*)$. While it is possible to formulate a continuous PEP with the performance measure as $f(X(T))-f(x^*)$ (as done in the discrete-time case in [26, Appendix D]), we decided to use $\tilde{f}(X(T))-\tilde{f}(x^*)$ as the performance measure because it makes the construction of the PEP kernel more natural. Another compelling reason is its relevance to the well-known convergence rates for the Triple Momentum Method (TMM) and the Information-Theoretic Exact Method (ITEM), we aim to analyze whose ODE models in our paper. The known Lyapunov functions for these methods (see [26, Section 2]) yields convergence rates for the quantity $f(x_{N})-f(x^*)-\frac{\mu}{2(1-\mu/L)}\|x_{N}-x^*\|^{2}$, not for $f(x_{N})-f(x^*)$. The continuous-time counterpart for this quantity is preicsely $\tilde{f}(X(T))-\tilde{f}(x^*)$.
>
> > Is it possible that the supremum ...
>
> In the literature on minimizing gradient norm of convex functions following the work [8], it is standard to assume the initial function condition (IFC), that is, $f(x_0)-f(x^*)$ is bounded by some constant. In our work, we assume that the supremum in the inequality (15) is finite, and this assumption can be considered as a natural extension of (IFC) to the strongly convex setting.
>
> > What is $x^{T}$ in equation (20) and (21)?
>
> $x^{T}:=X(T)$. We will clarify this in the final version.
>
> > Typos: line 45, 'enhances'; page 5, footnote 3 line 2, 'space'.
>
> Thank you for noting these typos. We will fix them in the final version.
>
> The reviewer has also provided valuable suggestions, which will be incorporated into the final version.

---

> > ### Comment · Reviewer_RgYv · 2023-08-16
> >
> > Thank you for your replies to my questions and updates to the manuscript. The primary concerns I had have been adequately addressed and clarified. As a result, I have adjusted my score to 5 accordingly.

---

### Official Review · Reviewer_C3yE · 2023-07-07

**Soundness:** 3 good
**Presentation:** 3 good
**Contribution:** 3 good
**Rating:** 7
**Confidence:** 4

**Summary:**

This paper provides a new technique (that is fundamentally different from the Lyapunov approach) for systemically analyzing the convergence rates of ODE models for first-order optimization methods, which reduces to verifying the positive semidefiniteness of specific Hilbert-Schmidt integral operators. This is a continuous time version of the Drori and Teboulle's (discrete) performance estimation problem (PEP) approach, which has recently become a fundamental tool in systemically analyzing the convergence rates of first-order methods. As a verification, the authors utilized the proposed tool for analyzing various ODE models of accelerated first-order methods such as (unified) AGM, TMM, ITEM, AGM-G and OGM-G, which either recovered the existing convergence rates or revealed new rates for various ODE models.

**Strengths:**

- This provides a new systematic analysis of convergence rates of the ODE models of first-order methods, which will be potentially useful for analyzing new ODE models. The authors have successfully verified the effectiveness of the tool for various ODE models.

- This continuous PEP resembles the discrete PEP, but its construction via the functional analysis is certainly not trivial and is very interesting.

- This provides new insights to the PEP analysis.

**Weaknesses:**

- One of the important features of the discrete PEP is that one can numerically find the values of Lagrange multipliers by numerically optimizing them, which can be used to reveal an analytical form of Lagrange multipliers. However, this paper's continuous PEP requires one to explicitly have the appropriate Lagrange multipliers, which can be laborious especially when dealing with new ODE models.

**Questions:**

- Line 132: How about letting the readers know that the derivation can be found in Appendix B?
- Line 180: Is this rate consistent with the rate in [9]?
- Appendix Line 75: $K^1$? $K^d$?

**Limitations:**

---

> ### Author Rebuttal · Authors · 2023-08-10
>
> We thank the reviewer for the positive feedback and thoughtful comments.
>
> > One of the important features of the discrete PEP is that one can numerically find the values of Lagrange multipliers by numerically optimizing them, which can be used to reveal an analytical form of Lagrange multipliers. However, this paper's continuous PEP requires one to explicitly have the appropriate Lagrange multipliers, which can be laborious especially when dealing with new ODE models.
>
> This concern does not significantly impact our contributions at this point. Our work primarily focuses on establishing a theoretical foundation, rather than dealing with implementation.
>
> Although it is not the main focus of our paper, the implementation of continuous PEP is a possible future direction. It would be necessitate to implement an optimization problem in function spaces, which might involve approximating continuous-time functions using a finite set of basis functions, with an appropriate care of numerical approximation errors.
>
> One can use the continuous PEP as a guiding framework for the discrete PEP. In this scenario, numerically optimizing Lagrange multiplier can be done in discrete PEP, and then taking the limit of stepsize to guess an explicit form of the working multiplier function for the continuous PEP.
>
> > Line 132: How about letting the readers know that the derivation can be found in Appendix B?
>
> Thank you for the suggestion. We will incorporate this into the final version of our paper.
>
> > Line 180: Is this rate consistent with the rate in [9]?
>
> In [9, Corollary 8], a convergence rate of $f(X(T))-f(x^*)\leq O(\mathrm{csch}^{2}(\frac{\sqrt{\mu}}{2}T))$ is reported (note that the notation $\mathrm{cschc}$ in [9] is defined as $\mathrm{cschc}(t):=t\mathrm{csch}(t))$. While our convergence guarantee in Line 180 is not exactly the same as the one in [9] because our one is for $\tilde{f}(X(T))-\tilde{f}(x^*)$, both rates are consistence in the sense that both exhibits the rate $O(\mathrm{csch}^{2}(\frac{\sqrt{\mu}}{2}T))$ . We will make this point clear in the final version of our paper.
>
> > Appendix Line 75: $K^{1}$? $K^{d}$?
>
> As $K^{d}$ is already defined in Appendix Line 63, $K^{1}$ is right here. However, there is a typo in Line 75: $L^{2}([0,T];\mathbb{R}^{d})$ should be corrected to $L^{2}([0,T];\mathbb{R}^{1})$. Thank you for the catch.

---

> > ### Comment · Reviewer_C3yE · 2023-08-17
> >
> > Thank you for the response. I have read the rebuttal and comments. I have no further questions and will keep my score.

---

### Official Review · Reviewer_pACc · 2023-07-17

**Soundness:** 3 good
**Presentation:** 3 good
**Contribution:** 3 good
**Rating:** 6
**Confidence:** 5

**Summary:**

Continuous counterpart of the work by Drori and Teboulle [2] was presented. Specifically, through the dual objective of the relaxed PEP in continuous time, the convergence rates of various ODEs were obtained. The analysis on the Lagrangian dual will lead to a dual solution based on a symmetric kernel of the Hilbert-Schmidt integral operator (Theorem 1).  Some of the follow up results were new to the literature e.g., low-resolution ODE (as opposed to high-resolution ODEs proposed by Shi et al. [19]) for the TM, ITEM methods & velocity/gradient norm convergence rates (Theorem 2).

**Strengths:**

1- The work has a solid mathematical base. The proofs are well written and mostly easy to follow.

2- The rate for TM ODE and ITEM ODE matching their discretized algorithms.

3- Investigating the connection between the continuous-time PEP and its discrete counterpart in Appendix G.2

**Weaknesses:**

1- The future directions are not clearly specified or very vague. This will impact the contribution of the work to the Neurips community. In the "Questions", I have asked some questions (numbers 2,3,4,5) which can be as future directions or even increase the contribution of this work beyond the current status.

2- The presentation can improve e.g.

-In (Relaxed PEP): move subject to under max for better readability.

-In 145: Sinse is a typo.

-References 22 23 are the same.

-Appendix: 455 missed parenthesis for 70.


3- Limited literature review. More comprehensive literature review could be placed in the Appnedix to save space.

4- Inconsistency in some parts of the text; e.g., in 113 I expected to see the inequalities mentioned in 112, but I was noted with another representation of AGM ODE from [9]. Also, sudden shifts from convex cases to strongly convex ones can cause confusion.

5- Some parts could be moved to the appendix (e.g. recovering the famous known rates for ODEs) and instead report their results in a table. Then, the authors could use the additional space to draw conclusions, compare (like graphs and visual simulations for instant comparison), or even exploring other aspects of the proposed framework or ODEs (by e.g. answering some of the questions in the "Questions" section below).

**Questions:**

1- Is the statement in 33[However ...]36 true? There has been a line of work which used positive-semidefiniteness to define tight Lyapunov functions (see [1]). This work however deals with a more general case which entails various ODEs through the choice of the kernel.



2- Following Q1, is it possible to use your framework to define tight Lyapunov functions? This relates to finding the analytical Lyapunov functions simulated in e.g. [2].



3- Is it possible to extend this work to the high-resolution ODE framework proposed by Shi to bypass the Lyapunov based proofs in their analysis?



4- Is there any connection between the low-resolution ODE for TMM proposed in this work and the high-resolution ODE proposed in [3]?



5- In authors' mind, what discretizations can recover the TMM and ITEM from the proposed TMM  and ITEM ODEs?


References
[1] Sanz-Serna, Jesús María and Konstantinos C. Zygalakis. “The connections between Lyapunov functions for some optimization algorithms and differential equations.” SIAM J. Numer. Anal. 59 (2020): 1542-1565.

[2] Upadhyaya, M., Banert, S., Taylor, A.B., & Giselsson, P. (2023). Automated tight Lyapunov analysis for first-order methods.

[3] B. Sun, J. George and S. Kia, "High-Resolution Modeling of the Fastest First-Order Optimization Method for Strongly Convex Functions," 2020 59th IEEE Conference on Decision and Control (CDC), Jeju, Korea (South), 2020, pp. 4237-4242, doi: 10.1109/CDC42340.2020.9304444.

**Limitations:**

The limitations are stated in the text.

---

> ### Author Rebuttal · Authors · 2023-08-09
>
> **Q2.** In Summary: Our continuous PEP is intrinsically associated with certain Lyapunov functions. However, if you are asking about the conventional Lyapunov function argument, where the Lyapunov function takes specific forms like
> $\mathcal{E}(t)=a(t)(f(X(t))-f(x^*)+b(t)\Vert Z(t)-x^{*}\Vert^{2},$
> then the answer is no.
>
> Let's see how our framework can be interpreted using a Lyapunov function. The continuous PEP presented in Sections 3.1 and 3.2 is related to the following Lyapunov function:
>
> $\mathcal{E}(t):=\nu\Vert x_0-x^*\Vert^2+\int_{0}^{t}\lambda_{1}(s)\left(\dot{\varphi}(s)+\left\langle \gamma(s),\int_{0}^{s}H(s,\tau)\gamma(\tau)\,d\tau\right\rangle \right)ds\Vert x_0-x^*\Vert^{2}$
>
> $\qquad\quad +\int_{0}^{t}\lambda_{2}(s)\left(\varphi(s)+\left\langle \gamma(s),v+\int_{0}^{s}\int_{\tau}^{s}H(\sigma,\tau)\,d\sigma\gamma(\tau)\,d\tau\right\rangle \right)dt\Vert x_{0}-x^{*}\Vert^{2},$
>
> which is decreasing by its construction. When the multiplier functions form a feasible solution to (7), i.e., $\lambda_{1}(0)=0$, $\lambda_1(T)=1$, and $\dot{\lambda}\_1(t)=\lambda_{2}(t)$, we have $\mathcal{E}(T)=f(X(T))-f(x^{*})+Q(T)$, where
>
> $\mathcal{Q}(T)=\left(\frac{1}{2}\langle K^{d}\gamma,\gamma\rangle+\langle\lambda_{2}(t)v,\gamma(t)\rangle+\nu\right)\|x_{0}-x^{*}\|^{2}.$
>
> If the PEP kernel (8) is positive semidefinite, we can show $Q(T)\geq0$, following the argument in Appendix Lines 73–78. As a result, we obtain
>
> $f(X(T))-f(x^*)=\mathcal{E}(T)-Q(T)\leq\mathcal{E}(T)\leq\mathcal{E}(0)=\nu\Vert x_0-x^*\Vert^{2}.$
>
> In the conventional Lyapunov function argument, we have an expression for $Q(T)$ that is automatically nonnegative (for example, as being a sum of squared norms). In contrast, the Lyapunov argument corresponding to PEP requires showing $Q(T)\geq0$ by using the positive semidefiniteness of the PEP kernel.
>
> **Q1.** As noted by the reviewer, there exists a line of work for systematically finding (conventional) Lyapunov functions. While we have cited some of these works in the related work section, we agree that mentioning this research direction around line 33 is appropriate. We will incorporate this in the final version of our paper.
>
> Nevertheless, the statement in lines 33–36 remains valid. Our framework circumvents the need for designing Lyapunov functions, resulting in a distinctive and more widely applicable way to establish convergence guarantees. In our framework, we can address all possible convergence rates obtained through a weighted integral of (5), with each corresponding to a dual feasible solution to (7). This capability enables us to show the optimality of convergence rates, as exemplified in Remark 1. Such a claim cannot be made straightforwardly in works focused on Lyapunov functions.
>
> **Q5.** TMM and ITEM can be expressed as the following fixed-step first-order method:
>
> $x_{i+1}=x_{i}-\frac{1}{L}\sum_{j=0}^{i}h_{ij}\nabla f(x_{j}).$
>
> The continuous-time counterpart of this form is the following dynamical system (4) (see [9, Section 2.4.2]):
>
> $\dot{X}(t)=-\int_{0}^{t}H(t,\tau)\nabla f(X(\tau))\,d\tau,$
>
> where $H(i\sqrt{s},j\sqrt{s})\approx h_{ij}$. In light of this, the discretization process can be understood as discretizing a kernel $H(t,\tau)$ into a matrix $[h_{ij}]$.
>
> **Q3.** It seems that such an extension might not be straightforward. The low-resolution ODEs take on the form $\ddot{X}(t)+b(t)\dot{X}(t)+c(t)\nabla f(X(t))=0$, which can be rewritten as (4) by Proposition 4 in the appendix. However, the high-resolution ODEs for Nesterov's AGM involve a gradient correction term $\nabla^{2}f(X)\dot{X}$, which cannot be derived from (4). To extend our continuous PEP framework to the high-resolution ODE framework, one should first find a reasonable generalization of the dynamics (4) that can handle the term $\nabla^{2}f(X)\dot{X}$, while enabling the convergence analysis via positive semidefinite kernels. This task is not trivial. Therefore, we defer it to future work.
>
> **Q4.** The high-resolution TMM ODE in [3] does not align with the low-resolution TMM ODE in our paper. This is due to the different choices of time step size. While we adopted a time stepsize of $1/\sqrt{L}$, [3] employed a time stepsize of $\sqrt{\alpha}$, where $\alpha=\frac{2-1/\sqrt{L/\mu}}{L}$ (the value of $\alpha$ depends on the specific algorithm). Although both choices make sense, our choice of stepsize $1/\sqrt{L}$ is more commonly employed in the literature and can be applied to any fixed-step first-order method $x_{i+1}=x_{i}-\frac{1}{L}\sum_{j=0}^{i}h_{ij}\nabla f(x_{j})$.
>
> **W5.** Thank you for providing a valuable suggestion. In the final version of our paper, we plan to include figures that visually illustrate PEP kernels (see Figures 1 and 2 in the attached pdf), along with detailed explanations. Additionally, we will enhance the conclusion section to offer a more comprehensive discussion of our framework and its potential applications in future research.
>
> **W1.** We will incorporate the reviewer's suggestions, as well as other possible future directions, e.g., obtaining optimal ODE model with the use of PEP.
>
> **W2.** Thank you for your suggestions; we will address these issues.
>
> **W3.** In our initial paper submission, we focused on including key papers related to our work, considering space limitations. Following your suggestion, we plan to utilize the appendix to provide a more comprehensive literature review. For instance, our literature review will cover papers related to PEP for finding tight Lyapunov functions and the generalization of PEP beyond the convex optimization setup.
>
> **W4.** Thank you for your suggestions. To improve readability, around line 113, we will first express equation (5) in a more familiar form:
>
> $0=\frac{d}{dt}( f(X(t))-f(x^*)) -\left\langle \nabla f(X(t)),\dot{X}(t)\right\rangle $
>
> $0\geq f(X(t))-f(x^*)-\left\langle \nabla f(X(t)),x^*-X(t)\right\rangle .$
>
> Also, before presenting Theorem 1, we will clearly mention that we consider the strongly convex case.

---

> > ### Comment · Reviewer_pACc · 2023-08-14
> > **Thank you for the response**
> >
> > I would like to thank the authors for their responses to my concerns and questions. I would like to add that regarding Q2 response it would be interesting to see if positive semi definiteness of (8) leads to structural constraints $Q(T)$. This might be an easy task to check for special cases like Nesterov's. Also, on author's response to Q5, I agree on their response and this is the systematic way of discretising any first-order method. However, in my understanding (and please correct me if I am wrong) the ODEs which were proposed for most of the methods in the paper (like the proposed TM method's ODE) are second-order ODEs and do not fit the class of ODE
> > $$\dot X(t)=-\int_0^{t}H(t,\tau)\nabla f(X(\tau))d\tau.$$
> >
> > My question was mainly regarding discretisers like explicit Euler with updated gradient calculation or semi-implicit Euler (SIE). For example in [1] it was shown that the NAG is the SIE discretization of a high-resolution ODE, but as far as I know, this is not the case for low-resolution ODEs (like yours) and more complicated discretizers are needed for these ODEs like rate-matching in [2]. Do you think one can find a discretiser like rate-matching to recover TM method from the low-resolution ODE you proposed?
> >
> > **References**
> >
> > [1] B. Shi, S. S. Du, W. J. Su, and M. I. Jordan. Acceleration via symplectic discretization of high- resolution differential equations (2019)
> >
> > [2] A. Wibisono, A. C. Wilson, and M. I. Jordan. A variational perspective on accelerated methods in optimization (2016)

---

> > > ### Author Response · Authors · 2023-08-15
> > >
> > > Thank you for your engagement in the discussion. We answer the questions and comments below.
> > >
> > > > However, in my understanding (and please correct me if I am wrong) the ODEs which were proposed for most of the methods in the paper (like the proposed TM method's ODE) are second-order ODEs and do not fit the class of ODE (4): $\dot{X}(t)=-\int_0^t H(t,\tau)\nabla f(X(\tau))d\tau$.
> > >
> > > We would like to clarify that all second-order ODEs indeed fall within the class of continuous-time dynamical systems of the form (4). In the appendix of our paper, Proposition 4 shows that a second-order ODE $\ddot{X}(t)+b(t)\dot{X}(t)+c(t)\nabla f(X(t))=0$ can be equivalently expressed as the integro-differential equation (4) with $H(t,\tau)=c(\tau)e^{-\int_{\tau}^{t}b(s)ds}$. For instance, the TM method (TMM)'s ODE $\ddot{X}+3\sqrt{\mu}\dot{X}+2\nabla f(X)=0$ can be transformed into the form (4), using $H(t,\tau)=2e^{-\int_{\tau}^{t}3\sqrt{\mu}ds}=2e^{3\sqrt{\mu}(\tau-t)}$.
> > >
> > > > I would like to add that regarding Q2 response it would be interesting to see if positive semi definiteness of (8) leads to structural constraints $Q(T)$.
> > >
> > > In our understanding, the answer is negative. It is a nontrivial task to obtain a Lyapunov function from the PEP kernel. In the conventional Lyapunov argument, we often have $\dot{\mathcal{E}}(t)<0$. Consequently, the expression of $\dot{\mathcal{E}}(t)$ as a quadratic functional of $\tau\mapsto\gamma(\tau)=\nabla f(X(\tau))$, i.e., the integral operator $K(t)$ for which $\dot{\mathcal{E}}(t)=-\langle\gamma,K(t)\gamma\rangle$, is involved in the PEP kernel. In fact, the PEP kernel $S$ can be obtained by integrating the kernel $K(t)$ over time and then adding the kernel associated with $Q(T)$, i.e., we have $S=\int K(t)dt+Q(T)$. However, by only knowing the PEP kernel $S$, one cannot determine the kernels $K(t)$ and $Q(T)$.
> > >
> > > It is worth noting that the converse of your statement is true. In order to show $S\succeq0$, one can first show $K(t)\succeq0$ and $Q(T)\succeq0$, which directly follows from the structure of $\dot{\mathcal{E}}(t)$ (for example, a squared distance), and then use the fact that a weighted integral of positive semidefinite kernels is positive semidefinite.
> > >
> > > > My question was mainly regarding discretisers like explicit Euler with updated gradient calculation or semi-implicit Euler (SIE). For example in [1] it was shown that the NAG is the SIE discretization of a high-resolution ODE, but as far as I know, this is not the case for low-resolution ODEs (like yours) and more complicated discretizers are needed for these ODEs like rate-matching in [2]. Do you think one can find a discretiser like rate-matching to recover TM method from the low-resolution ODE you proposed?
> > >
> > > While this question seems tangential to our contributions, we can provide an answer.
> > >
> > > Shi et al. (2019) showed that AGM-SC differs from the semi-implicit Euler scheme applied to the high-resolution AGM-SC ODE by only a factor of $\frac{1}{1-\sqrt{\mu s}}$. As the reviewer correctly noted, applying this discretization technique to low-resolution ODEs does not yield Nesterov's AGM. The reason is that the low-resolution ODEs do not capture the gradient descent step $x_{k}=y_{k-1}-s\nabla f(y_{k-1})$ in AGM. Thus, when discretizing low-resolution ODEs, it is essential to incorporate the gradient step into the naive discretization scheme.
> > >
> > > The derivation of TMM ODE from TMM is provided in Appendix D.1. Reversing this procedure gives the methodology for discretizing TMM ODE. Applying the explicit Euler method to $\dot{Z}=\sqrt{\mu}(Y-Z-\frac{1}{\mu}\nabla f(Y))$ gives $z_{k+1}-z_{k}=\sqrt{\mu s}(y_{k}-z_{k}-\frac{1}{\mu}\nabla f(y_{k}))$. Applying the implicit Euler method to $\dot{Y}=2\sqrt{\mu}(Z-Y)$ gives $y_{k}-x_{k}=2\sqrt{\mu s}(z_{k}-x_{k})$, where $y_{k-1}$ is replaced by $x_{k}$. By incorporating the gradient step $x_{k}=y_{k-1}-s\nabla f(y_{k-1})$ and adjusting the coefficient $2\sqrt{\mu s}$ to $\frac{2\sqrt{\mu s}}{1+\sqrt{\mu s}}$, we recover TMM.
> > >
> > > ---
> > >
> > > Please feel free to ask for further clarifications.

---

> > > > ### Comment · Reviewer_pACc · 2023-08-15
> > > > **Official Comment by reviewer**
> > > >
> > > > Thank you for your clarifications.

---

### Author Rebuttal · Authors · 2023-08-10

Dear all reviewers,

# Figures

The attached PDF file contains the figures mentioned in the rebuttals: Visualization of PEP kernels, and numerical experiment for the convergence rate of AGM-SC ODE obtained in Section 4.

# Relaxing assumptions in Theorem 2

We have relaxed the assumption (16) in Theorem 2. This modification will not affect other parts of our paper, as the revised version is a generalization of the initial version.

## Theorem statement

Remove the assumption (16) and replace the transformation before line 226 with the following:

$g^{G}(t)=\lambda(t)g(t)-\int_{0}^{t}\dot{\lambda}(\tau)\nabla g(\tau)d\tau,$

where $g(t)=\nabla f(X(t))+\mu\int_{0}^{t}\int_{\tau}^{t}h^{G}(s,\tau)ds\nabla f(X(\tau))d\tau$.

(Note: we have shown that this transformation is equivalent to that in the initial submission.)

## Proof

We make the following modifications on Lines 103--117.

Remove Lines 103--108, 110.

Let $\bar{f}(x)=f(x)-\frac{\mu}{2}\|x-x_{0}\|^{2}$. Then, $\bar{f}$ is convex.

Change the definition of $\tilde{f}_{t}(y)$ as follows:

$ \tilde{f}_t(y)=\lambda(t)(\bar{f}(y)-\bar{f}(X(T)))$

$\qquad \qquad -\langle \int_0^t\dot{\lambda}(\tau)\nabla\bar{f}(X(\tau))d\tau,y-X(T)\rangle . $

Change the formula after Line 112 as follows:

$ \frac{\partial}{\partial t}\tilde{f}\_{t}(y)|_{y=X(t)} =\dot{\lambda}(T)\left(\bar{f}(X(t))-\bar{f}(X(T))-\left\langle \nabla\bar{f}(X(t)),X(t)-X(T)\right\rangle \right). $

Change the definition of $N(t)$ as follows:

$N(t)=\frac{1}{M}\left(\bar{f}(y)-\bar{f}(X(T))-\left\langle \nabla\bar{f}(X(t)),y-X(T)\right\rangle \right).$

---

### Decision · Program_Chairs · 2023-09-21

**Decision:**

Accept (poster)

**Comment:**

The paper introduces a continuous time counterpart to the performance estimation problems (PEP) of Drori and Teboulle, offering a new systematic analysis of convergence rates of the ODE models of first-order methods. The reviewing team unanimously agreed that the proposed methodology is interesting and the paper satisfies the acceptance criteria. The paper's technical foundation is found to be solid. While the approach builds on PEP, designing the continuous-time PEP through functional analysis presents a nontrivial challenge. The article is presented well with numerous examples that demonstrate the effectiveness of the method.

In terms of weaknesses, the paper lacks experimental evaluation since its focus is primarily on the theoretical foundation of continuous-time PEP. The implementation is deferred to future work, though a basic numerical experiment for the convergence rate of AGM-SC ODE is provided in the rebuttal. Some reviewers raised concerns about the difficulty of identifying suitable Lagrange multiplier functions, which can be considered as a potential drawback. Although no systematic way to find suitable multiplier functions is presented, the authors suggest in their rebuttal a rule of thumb for selecting appropriate functions when we have a prediction on the final convergence rate. A reviewer suggested expanding the literature review. The authors are encouraged to incorporate these suggestions in the camera-ready version and to provide detailed explanations addressing these concerns where relevant.